# Operational analysis for COVID-19 testing: Determining the risk from asymptomatic infections

**Marc Mangel** [1,2,3]*

**1** Department of Biology, University of Bergen, Bergen, Norway, **2** Department of Applied Mathematics, University of California Santa Cruz, Santa Cruz, CA, United States of America, **3** Puget Sound Institute, University of Washington Tacoma, Tacoma, WA, United States of America

* msmangel@ucsc.edu

**Data Availability Statement:** All relevant data are within the paper and its Supporting information files.

**Funding:** MM was supported by a consulting contract with the Johns Hopkins University Applied

## Abstract

Testing remains a key tool for managing health care and making health policy during the coronavirus pandemic, and it will probably be important in future pandemics. Because of false negative and false positive tests, the observed fraction of positive tests—the surface positivity—is generally different from the fraction of infected individuals (the incidence rate of the disease). In this paper a previous method for translating surface positivity to a point estimate for incidence rate, then to an appropriate range of values for the incidence rate consistent with the model and data (the test range), and finally to the risk (the probability of including one infected individual) associated with groups of different sizes is illustrated. The method is then extended to include asymptomatic infections. To do so, the process of testing is modeled using both analysis and Monte Carlo simulation. Doing so shows that it is possible to determine point estimates for the fraction of infected and symptomatic individuals, the fraction of uninfected and symptomatic individuals, and the ratio of infected asymptomatic individuals to infected symptomatic individuals. Inclusion of symptom status generalizes the test range from an interval to a region in the plane determined by the incidence rate and the ratio of asymptomatic to symptomatic infections; likelihood methods can be used to determine the contour of the rest region. Points on this contour can be used to compute the risk (defined as the probability of including one asymptomatic infected individual) in groups of different sizes. These results have operational implications that include: positivity rate is not incidence rate; symptom status at testing can provide valuable information about asymptomatic infections; collecting information on time since putative virus exposure at testing is valuable for determining point estimates and test ranges; risk is a graded (rather than binary) function of group size; and because the information provided by testing becomes more accurate with more tests but at a decreasing rate, it is possible to over-test fixed spatial regions. The paper concludes with limitations of the method and directions for future work.

Physics Laboratory. The funders had no role in study design, data collection and analysis, decision to publish, or preparation of the manuscript.

**Competing interests:** The author has declared that no competing interests exist.

## Introduction

Entering the third year of the 2019 coronavirus disease (henceforth COVID-19) pandemic, it is clear that the world was woefully underprepared, in many different ways, for dealing with it. ItThe current pandemic is an illustration of the natural evolutionary play [1] so that one should expect future pandemics and lose no time in preparing for them while dealing with the present one.

Some authors have suggested that it is appropriate to prepare for the next pandemic as one prepares for war [2–5]. Operational analysis grew out of the scientific approach to operational questions in World War II [6–9]. One of the key tenets of operational analysis is to model the process as well as the data [7, 10]. Among the advantages, when the process is modeled, one knows the true state of world, which allows assessment of the quality of the analyses by comparison of analytical outcomes with a known situation. This gives confidence that the methods will work when the true state of the world is unknown.

Models for dynamics and control of the disease, prioritizing hospital care, and setting policy [11–18] require information about the health status of the population. This is determined by testing for infection, which thus emerged as a crucial component of managing health policy during the current pandemic and will probably be key in future pandemics [19]. For example, the Global Influenza Surveillance and Response Network (the "flu network" [20, 21]) established in 1952 played a key role in the early responses to the COVID-19 pandemic. The time is now to prepare for future testing.

Testing is complicated an individual is in an early stage of the infection may give a false negative test, infected individuals may be asymptomatic and thus not tested, and symptomatic but uninfected individuals may give false positive test results. Thus, test errors involve both false negative tests, and false positive tests, in which an uninfected individual tests positive [22–27]. These are called the false negative probability [28], denoted here by $p_{FN}$, and false positive probability, denoted here $p_{FP}$. It may one day be possible to drive the false positive probability to zero with improved specificity of tests, but the ontogeny of the disease within an individual means that there will always be false negative tests [25, 26].

A starting point for the interpretation of testing results is to envision that a population is divided into infected (antigen positive) and uninfected (antigen negative) individuals, with the goal of estimating the fraction of infected individuals (the incidence rate) from the number of positive results $P$ when $T$ tests are given. Because of both kinds of test errors, the surface positivity rate $P/T$ (which is observed; henceforth simply called positivity rate) will generally differ from the incidence rate (which is not observed). It is natural and intuitive to ask for the unobserved incidence rate that is most likely given the test results; this is called the Maximum Likelihood Estimate (MLE) of the incidence rate.

Brown and Mangel [29] and Mangel and Brown [30] (also see [31, 32] where similar methods are used) show that the Maximum Likelihood Estimtate for the incidence rate, denoted by $\hat{f}$ is

$$\hat{f} = \frac{P/T - p_{FP}}{1 - p_{FN} - p_{FP}},\qquad(1)$$

which is to be interpreted as $\hat{f} = 0$ if the right side of Eq 1 is negative. As will be explained below application of Bayesian methods allows determination of a probability distribution for the incidence rate when the right side of Eq 1 is negative.

In addition to a point estimate for the incidence rate, it is valuable to have a range of incidence rates that are consistent with the model and the data since then one can bound the incidence rate and its associated risk of further infection (Eqs 3 and 4 below). That is, forecasting

for a pandemic can be improved by using a predictive distribution, rather than the point estimates in Eq 1, (cf. [33]).

In [29, 30], we show that an appropriate test range, denoted by $Range(\hat{f})$ is

$$Range(\hat{f}) = 3.92\sqrt{\frac{p_+(\hat{f})(1 - p_+(\hat{f}))}{T(1 - p_{FN} - p_{FP})^2}}, \tag{2}$$

where $p_+(\hat{f}) = \hat{f}(1 - p_{FN}) + (1 - \hat{f})p_{FP}$.

McElreath [34, p. 54] describes and equation such as Eq 2 as the 95% compatibility interval, avoiding the undesired implications of words such as "confidence" or "credible" [35]. $Range(\hat{f})$ is symmetrically distributed around the true range with very small mean error between the two [30], so that lower and upper limits for the estimated incidence rate are $\hat{f}_{lower} = \hat{f} - 0.5 \cdot Range(\hat{f})$ and $\hat{f}_{upper} = \hat{f} + 0.5 \cdot Range(\hat{f})$.

Mangel and Brown [30] also show how likelihood methods can be used to obtain a test range when positivity is 0 (so that $\hat{f} = 0$). In this paper, we will show how to determine the test range when $0 < P/T < p_{FP}$ However, for the remainder of this section, we assume that $P/T > p_{FP}$ so that the estimate of incidence rate is strictly positive.

Eqs 1 and 2 lead to the operational recommendation that one should stratify testing data according to test errors. When this is not possible, one should stratify tests according to the estimated time since exposure, assign best estimates to the test errors, and conduct sensitivity analyses of the results.

Test results can play an important role in policymaking because they can be used to determine the risk of spreading infection associated with groups of different sizes. Doing so requires a definition of risk. We define the risk to be including at least one infected individual in a group of specified size. The risk $\mathcal{R}(h, \hat{f})$ associated with a group of size $h$ when the estimate for incidence rate is $\hat{f}$ is [29, 30]

$$\mathcal{R}(h, \hat{f}) = 1 - (1 - \hat{f})^h. \tag{3}$$

Eq 3 allows us to explore the risk ramifications of groups of different sizes. Were the true incidence rate known, we replace $\hat{f}$ by $f_t$ (Fig 1). Fig 1 can be used to determine the risk associated with groups of different sizes by choosing a group size on the x-axis, drawing a vertical line to intersect the curve, drawing a horizontal line that intersects the y-axis, and reading off the level of risk. When the true value of the incidence rate is unknown, we create upper and lower bounds for risk by generating curves similar to Fig 1 using the lower and upper bounds $\hat{f}_{lower} = \hat{f} - 0.5 \cdot Range(\hat{f})$ and $\hat{f}_{upper} = \hat{f} + 0.5 \cdot Range(\hat{f})$, as in Brown and Mangel [[29, Fig 2].

We can invert Eq 3 by specifying a level of acceptable risk $\mathcal{R}_{acc}$ and then solve fof the group size $h_{acc}(\mathcal{R}_{acc}, \hat{f})$ consistent with the specified acceptable risk and the estimate of incidence rate is $\hat{f}$:

$$h_{acc}(\mathcal{R}_{acc}, \hat{f}) = \frac{log(1 - \mathcal{R}_{acc})}{log(1 - \hat{f})}. \tag{4}$$

Replacing the estimate of incidence rate by its maximum and minimum values $f_{max} = \hat{f} + 0.5 \cdot Range(\hat{f})$ and $f_{min} = \hat{f} - 0.5 \cdot Range(\hat{f})$ in Eq 4 allows us to bound the acceptable group size consistent with the level of acceptable risk.

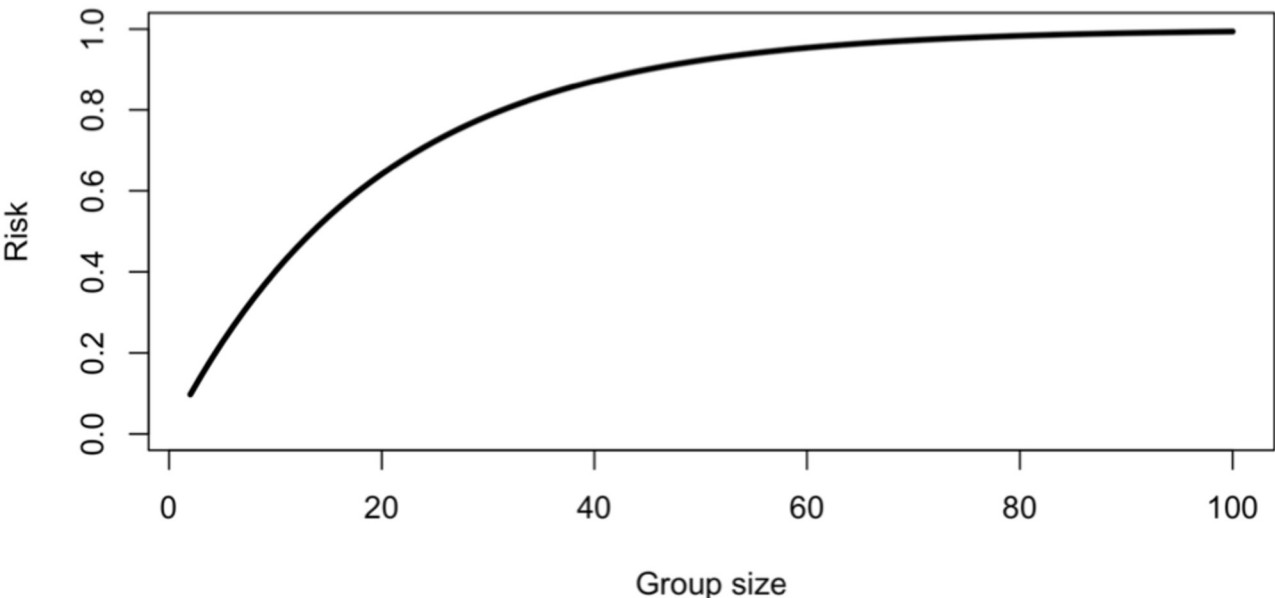

**Fig 1. The risk of groups of different sizes (Eq 3) when the true fraction of infected individuals is $f_t$ = 0.05 (i.e., we set $\hat{f} = 0.05$ in Eq 3).** This figure can be used to determine the risk associated with groups of different sizes (ranging from 2 to 100) by choosing a group size on the x-axis, drawing a vertical line to intersect the curve, drawing a horizontal line that intersects the y-axis, and reading off the level of risk.

In Fig 2, I show 16 realizations of acceptable group size using the simulation methods described in [30]. The dotted line shows the group size consistent with the level of acceptable risk when the incidence rate is $f_t$, the solid black line is the group size using the estimate in Eq 4, and the red and blue lines are the group sizes using the maximum and minimum estimates for incidence rate, $f_{upper} = \hat{f} + 0.5 \cdot Range(\hat{f})$ and $f_{lower} = \hat{f} - 0.5 \cdot Range(\hat{f})$, respectively. One key observation is that the group size determined if the incidence rate were known (dotted line) falls between those determined from the upper and lower limits of incidence rate determined by the test range.

It is also now well established that asymptomatic infected individuals can readily transmit infection [36–47]. Birx [2] emphasizes the role of untested asymptomatic individuals in the spread of the disease. Because of asymptomatic cases, policies that exclude symptomatic individuals from groups may still have considerable risk of including infected individuals who can transmit the disease.

The first purpose of this paper is to show how to obtain the test range when there is no information on symptoms and positivity is less than the probability of a false positive test. The second purpose of this paper is to generalize Eqs 1–4 and develop the analogue of Fig 1 when asymptomatic and symptomatic individuals are identified at the time of testing.

When there is information on symptoms (Fig 3), a fraction $f_t$ of the population is infected and symptomatic; a fraction $\rho_t f_t$ is infected and asymptomatic; a fraction $g_t$ is uninfected but symptomatic; and the remaining fraction, $1 - f_t(1 + \rho_t) - g_t$, is neither infected nor symptomatic. Infected individuals have probabilities of a false negative test, denoted by $p_{SFN}$ and $p_{AFN}$ where the subscript $S$ and $A$ correspond to symptomatic and asymptomatic individuals, respectively. Uninfected individuals have probabilities of a false positive test denoted by $p_{SFP}$ and $p_{AFP}$, respectively. Using testing information, we seek point estimates and the analogue of test ranges for the unknown incidence rates and ratio of asymptomatic to symptomatic cases.

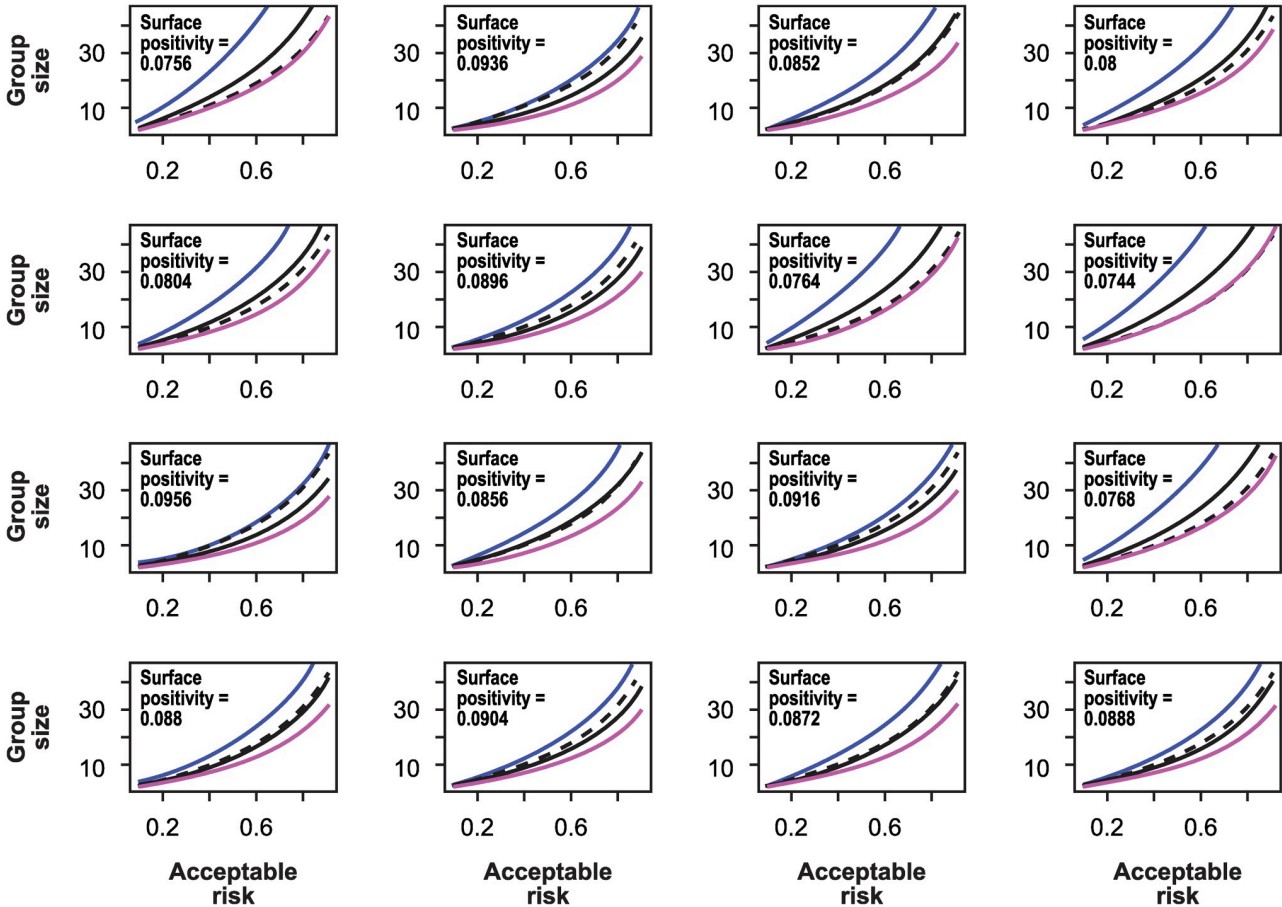

**Fig 2. Sixteen realizations of the group size as a function of acceptable risk using the simulation methods described in [30].** In all panels, the number of tests is $T = 2500$, the true incidence rate is $f_t = 0.05$, and the probabilities of false negative and false positives tests are 0.25 and 0.05. The dotted line shows the group size consistent with the level of acceptable risk when the incidence rate is $f_t$, the solid black line is the group size using the estimate in Eq 1 determined using the positivity rate from the individual realization of the simulation, and the red and blue lines are the group sizes using the maximum and minimum estimates for incidence rate, $f_{upper} = \hat{f} + 0.5 \cdot Range(\hat{f})$ and $f_{lower} = \hat{f} - 0.5 \cdot Range(\hat{f})$, respectively. One key observation is that the group size determined if the incidence rate were known (dotted line) falls between those determined from the upper and lower limits of incidence rate determined by the test range.

## Materials and methods

### Determining test range with no information on symptoms and positivity less than the probability of a false positive test

In this case, as with Eqs 1–4, there is a single unknown incidence rate, which we continue to denote by $f_t$. The methods used are generalized when there is information on symptoms, so this section is a warm-up to the harder problem.

The probability of obtaining a positive test when the incidence rate is $f$ is

$$p_+(f) = f(1 - p_{FN}) + (1 - f)p_{FP}. \tag{5}$$

The first term on the right hand side of Eq 5 corresponds to individuals who are infected and have a true positive test; the second term corresponds to individuals who are not infected and have a false positive test.

**Antigen positive?**

**Fig 3. The population divided into four classes according to infection and symptom status.** A fraction $f_t$ of the population is symptomatic and infected (antigen positive); such individuals have a probability of a false negative test $p_{SFN}$. A fraction $g_t$ of the population is symptomatic but not infected; such individuals have a probability of a false positive test $p_{SFP}$. A fraction $\rho_t f_t$ of the population is infected but not symptomatic; such individuals have a probability of a false negative test $p_{AFN}$. Finally, fraction $1 - f_t - g_t - \rho_t f_t = 1 - f_t(1 + \rho_t) - g_t$ of the population is neither infected nor symptomatic; such individuals have a probability of a false positive test $p_{AFP}$. The subscript $t$ indicates that these three parameters characterize the true state of the world, however none of them are observable.

When $T$ tests are given, the number of positive tests $P$ is binomially distributed with parameters $T$ and $p_+(f)$ [30–32], which we write as $P = \mathcal{B}(\cdot, T, p_+(f_t))$, where $\mathcal{B}(P, T, p_+(f)) = \binom{T}{P} p_+(f)^P (1 - p_+(f))^{T-P}$. The likelihood of an incidence rate $f$ given the test data $P$ and $T$ has the same form [30–32], but is a function of the incidence rate $f$ conditioned on the values of the test data

$$\mathcal{L}(p_+(f)|P, T) = \binom{T}{P} p_+(f)^P (1 - p_+(f))^{T-P}. \tag{6}$$

In S1 Section in S1 File, we show that the maximum likelihood estimate for the fraction of the population infected $\hat{f}$ satisfies

$$\frac{p'_+(\hat{f})}{p_+(\hat{f})(1 - p_+(\hat{f}))} [P - T p_+(\hat{f})] = 0 \tag{7}$$

where $p'_+(\hat{f}) = 1 - p_{FN} - p_{FP}$.

Since $p_+(\hat{f}) = \hat{f}(1 - p_{FN}) + (1 - \hat{f})p_{FP}$, we conclude that if there is an internal maximum of the likelihood (i.e. $\hat{f} > 0$), it must occur when $P = T p_+(\hat{f}) = T[\hat{f}(1 - p_{FN}) + (1 - \hat{f})p_{FP}]$. Solving this equation for $\hat{f}$ gives Eq 1. When $P \leq T p_+(f)$, we set $\hat{f} = 0$ and arrive at the nettlesome case of this subsection.

In Fig 4, I show the logarithm of the likelihood (the log-likelihood function) as a function of incidence rate $f$ for four values of positivity. In Fig 4(A), $P/T = 0.075$ and the peak of the likelihood is clearly away from the boundary $f = 0$. As positivity declines but stays larger than $p_{FP}$, as in Fig 4(B) and Fig 4(C), there is still an internal peak of the likelihood function. However,

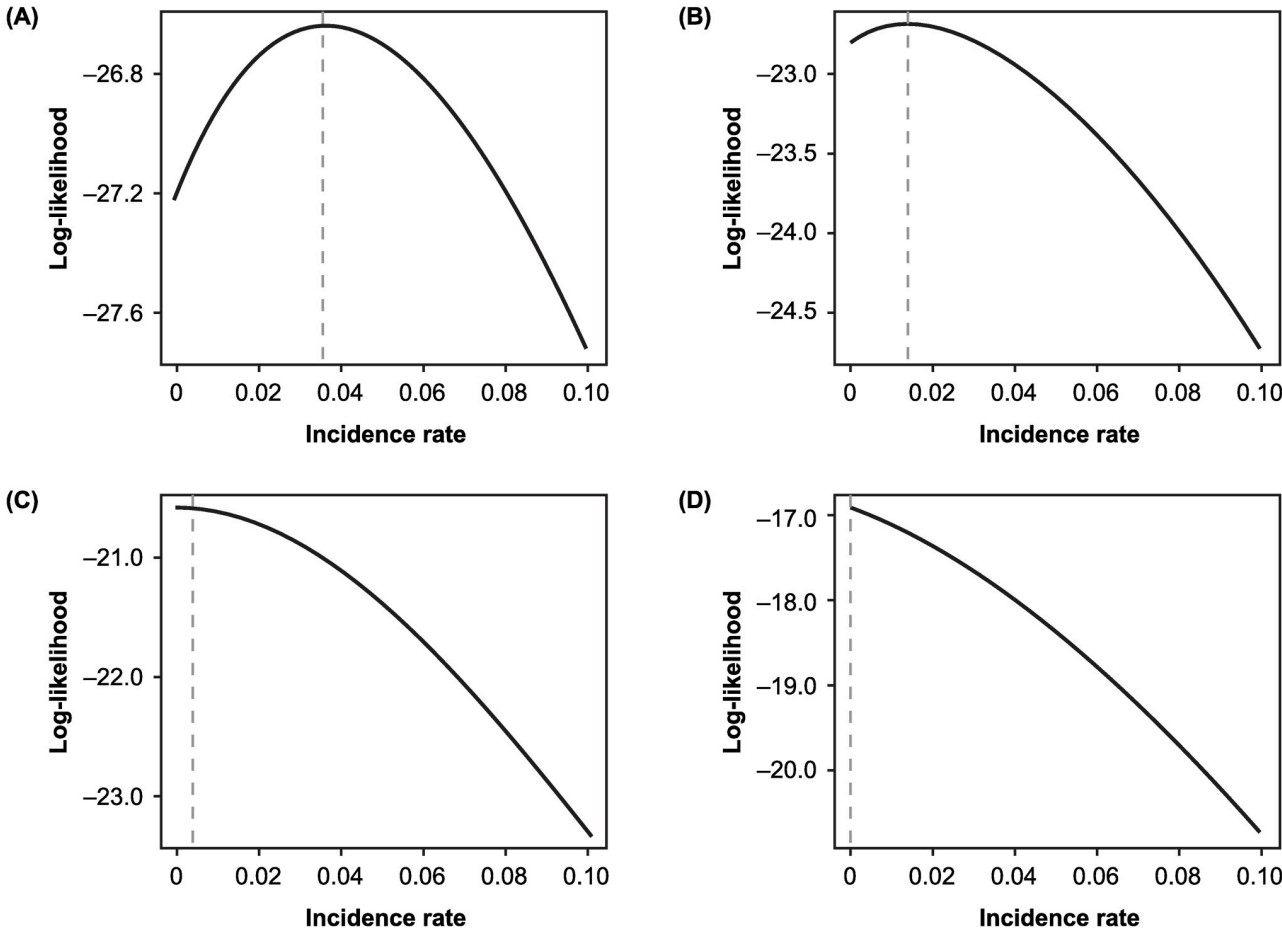

**Fig 4. Behavior of the log-likelihood function.** Shown is the log-likelihood function (the logarithm of the right side of Eq 6) as the positivity rate declines when $p_{FN} = 0.25$, $p_{FP} = 0.05$ and $T = 100$ tests are administered for positivity (A) 0.075, (B) 0.06, (C) 0.0525, and (D) 0.04.

when positivity falls below $p_{FP}$, as in Fig 4(D), the maximum of the log-likelihood function occurs on the boundary.

We convert from a likelihood to a probability distribution by assuming a uniform prior on $f$ and use Bayes's theorem to write the probability density for $f$ given the test data (also see S2 Section in S1 File):

$$\varphi(f|P, T) = \frac{\mathcal{L}(f|P, T)}{\int_{f'=0}^{1} \mathcal{L}(f'|P, T)df'}.\tag{8}$$

Although the denominator in Eq 8 can be written in terms of the classical beta function [48], it is most simply viewed as a constant obtained by using a very fine discretization of the interval [0, 1].

When the maximum of the likelihood occurs at the boundary $f = 0$, the probability $\varphi(f)$ will also have its maximum at the boundary. In this case, the test range is no longer symmetrical but is an interval $[0, f_{0.95}]$, where $f_{0.95}$ is the value of incidence rate such that $\int_{f'=0}^{f_{0.95}} \phi(f'|P, T)df' = 0.95$ (or the equivalent when a summation instead of an integral is used in Eq 8).

## Analysis when there is information on symptoms

**The operational situation with information on symptoms.** We assume that $T$ tests are administered to a population in which some individuals are symptomatic and others are not (recorded at the time of testing) and each individual tested has either a positive or negative test for coronavirus. As described in Fig 3, there are now four classes of individuals:

- A fraction $f_t$ of the population is symptomatic and infected (antigen positive); these individuals have a probability of a false negative test $p_{SFN}$.

- A fraction $g_t$ of the population is symptomatic but not infected; these individuals have a probability of a false positive test $p_{SFP}$.

- A fraction $\rho_t f_t$ of the population is infected but not symptomatic; these individuals have a probability of a false negative test $p_{AFN}$.

- The remaining fraction of the population, $1 - f_t - g_t - \rho_t f_t = 1 - f_t(1 + \rho_t) - g_t$, is neither infected nor symptomatic; these individuals have a probability of a false positive test $p_{AFP}$.

When this situation holds, three kinds of test data are generated:

- The number $P$ of positive tests.

- The number $T_S$ of symptomatic individuals.

- The number $P_S$ of symptomatic individuals who tested positive.

**Point estimates for the fractions of infected and uninfected symptomatic individuals and the ratio of asymptomatic to symptomatic infected individuals.** The following causal chain characterizes the operation of testing:

- The total number of tests, $T$, leads to number of symptomatic individuals in the sample, $T_S$.

- $T_S$ leads to the number of positive tests of symptomatic individuals, $P_S$.

- $T$, $T_S$, and $P_S$ combined lead to the remaining positive results, $P - P_S$ of $T - T_S$ tests from asymptomatic individuals.

As above, we let $\mathcal{B}(\cdot | N, p)$ denote a binomial distribution with number of samples $N$ and probability of a positive event $p$, where the dot can run from 0 (no positive event) to $N$ (only positive events). If $p_S$ denotes the probability of sampling a symptomatic individual, $p_{S+}$ the probability of obtaining a positive test from a symptomatic individual, and $p_{A+}$ the probability of obtaining a positive test from an asymptomatic individual, the test results have distributions

$$T_S \sim \mathcal{B}(\cdot | T, p_S), \tag{9}$$

$$P_S \sim \mathcal{B}(\cdot | T, T_S, p_{S+}), \text{ and} \tag{10}$$

$$P - P_S \sim \mathcal{B}(\cdot | T, T_S, P_S, p_{A+}). \tag{11}$$

The probabilities on the right sides of Eqs 9–11 are constructed from the assumptions summarized in Fig 3. The fraction of individuals who are symptomatic is $f_t + g_t$ so that

$$p_S = f_t + g_t. \tag{12}$$

Since $p_{S+}$ is the probability that an individual tests positive given that the individual is symptomatic, from the definition of conditional probability

$$p_{S+} = \frac{Pr[\text{symptomatic and test positive}]}{Pr[\text{symptomatic}]} = \frac{f_t(1 - p_{SFN}) + g_t p_{SFP}}{f_t + g_t}. \tag{13}$$

The probability that an individual tests positive given that the individual is asymptomatic is computed similarly:

$$p_{A+} = \frac{f_t \rho_t (1 - p_{AFN}) + (1 - g_t - f_t(1 + \rho_t)) p_{AFP}}{1 - f_t - g_t}. \tag{14}$$

We let $\hat{f}, \hat{g}$ and $\hat{\rho}$ denote the maximum likelihood estimates (MLEs) for the fraction of the population that is infected and symptomatic or not infected and symptomatic respectively, and for the ratio of the fraction that is infected and asymptomatic to that which is infected and symptomatic.

When a random variable has the binomial distribution $\mathcal{B}(\cdot | N, p)$, given $K$ positive events, the MLE for $p$ is $\hat{p} = K/N$ (see S1 Section in S1 File) so that the MLEs for the probabilities in Eqs 9–11 are $T_S/T$, $P_S/T_S$, and $(P - P_S)/(T - T_S)$ from which we conclude

$$\hat{f} + \hat{g} = \frac{T_S}{T}, \tag{15}$$

$$\frac{\hat{f}(1 - p_{SFN}) + \hat{g} p_{SFP}}{\hat{f} + \hat{g}} = \frac{P_S}{T_S}, \quad \text{and} \tag{16}$$

$$\hat{f}\hat{\rho}(1 - p_{AFN}) + (1 - \hat{f}(1 + \hat{\rho}) - \hat{g}) p_{AFP} = (1 - \hat{f} - \hat{g}) \frac{P - P_S}{T - T_S}. \tag{17}$$

Eqs 15 and 16 are independent of $\hat{\rho}$ and can be rewritten as

$$\hat{f} = \hat{g} \frac{(P_S/T_S) - p_{SFP}}{1 - p_{SFN} - (P_S/T_S)}, \tag{18}$$

which we write as $\hat{f} = c_1(P_S, T_S)\hat{g}$, where $c_1(P_S, T_S)$ is the combination of terms multiplying $\hat{g}$ on the right side of Eq 18 and the test errors are suppressed. With this notation, we substitute into Eq 15 and solve to obtain

$$\hat{g} = \frac{T_S}{(1 + c_1(P_S, T_S))T}. \tag{19}$$

Thus, both $\hat{f}$ and $\hat{g}$ are known; they are random variables because $P_S$ and $T_S$ are random variables.

We now rewrite Eq 17 as

$$\hat{f}\hat{\rho}(1 - p_{AFN} - p_{AFP}) = (1 - \hat{f} - \hat{g}) \frac{P - P_S}{T - T_S} - p_{AFP}(1 - \hat{f} - \hat{g}), \tag{20}$$

let $c_2 = 1 - p_{AFN} - p_{AFP}$, and solve for $\hat{\rho}$ to obtain

$$\hat{\rho} = \frac{(1 - \hat{f} - \hat{g})}{c_2 \hat{f}} \left[ \frac{P - P_S}{T - T_S} - p_{AFP} \right]. \tag{21}$$

Since the right side of Eq 21 depends on the test data $T_S$, $P_S$, and $P$, $\hat{\rho}$ is also a random variable.

Eqs 18, 19 and 21 generalize Eq 2 to the case in which symptomatic and asymptomatic individuals are identified at the time of testing. We have already thus generalized the method in [29, 30] to obtain point estimates of the fractions of the population of infected and symptomatic, uninfected and symptomatic, and infected and asymptomatic individuals. We next explore the properties of these point estimates and then generalize the notion of test range and compute and the risk of groups of different sizes including asymptomatic infected individuals.

**The means of the estimates.** We compute the means of the estimates, continuing to use $f_t$, $g_t$, and $\rho_t$ to denote their true values, with two goals. We explore 1) whether the estimates in Eqs 18, 19 and 21 are unbiased, in the sense that their expectations (over the stochastic sampling process) are the underlying true values generating the data, and 2) if there is a bias how to characterize it.

**The mean of $\hat{g}$.** We begin by rewriting Eq 19 as

$$\hat{g} = \frac{T_S}{T} \cdot \frac{1}{1 + c(P_S, T_S)}. \tag{22}$$

In S2 Section in S1 File, I show that $1 + c(P_S, T_S) = \frac{1 - p_{SFN} - p_{SFP}}{1 - p_{SFN} - (P_S / T_S)}$ so that we can rewrite Eq 22 as

$$\hat{g} = \frac{T_S}{T} \cdot \left[ \frac{1 - p_{SFN} - (P_S / T_S)}{1 - p_{SFN} - p_{SFP}} \right] = \frac{1}{T} \cdot \left[ \frac{T_S(1 - p_{SFN}) - P_S}{1 - p_{SFN} - p_{SFP}} \right]. \tag{23}$$

Since the denominator in Eq 23 is a constant, the expectation of $\hat{g}$ is

$$\mathcal{E}(\hat{g}) = \frac{1}{T} \cdot \frac{1}{1 - p_{SFN} - p_{SFP}} \cdot [\mathcal{E}(T_S)(1 - p_{SFN}) - \mathcal{E}(P_S)]. \tag{24}$$

In the S2 Section in S1 File, I show that $\mathcal{E}(T_S) = T(f_t + g_t)$ and $\mathcal{E}(P_S) = T[f_t(1 - p_{SFN}) + g_t p_{SFP}]$, from which it follows that $\mathcal{E}(\hat{g}) = g_t$; the expected value of $\hat{g}$ is the true value that underlies the testing process.

**The mean of $\hat{f}$.** We begin by multiplying the top and bottom of the right side of Eq 18 by $T_S$ to obtain

$$\hat{f} = \hat{g} \frac{P_S - p_{SFP} T_S}{T_S(1 - p_{SFN}) - P_S} \tag{25}$$

and now use the version of $\hat{g}$ on the far right side of Eq 23 to obtain

$$\hat{f} = \frac{1}{T} \cdot \left[ \frac{T_S(1 - p_{SFN}) - P_S}{1 - p_{SFN} - p_{SFP}} \right] \cdot \left[ \frac{P_S - p_{SFP} T_S}{T_S(1 - p_{SFN}) - P_S} \right] = \frac{1}{T} \left[ \frac{P_S - p_{SFP} T_S}{1 - p_{SFN} - p_{SFP}} \right]. \tag{26}$$

Taking expectations on the far right side of Eq 26, we obtain

$$
\begin{aligned}
\mathcal{E}(\hat{f}) &= \frac{1}{T} \cdot \frac{1}{1 - p_{SFN} - p_{SFP}} \cdot [T(f_t(1 - p_{SFN}) + g_t p_{SFP}) - T(f_t + g_t) p_{SFP}] \\
&= \frac{1}{T} \cdot \frac{1}{1 - p_{SFN} - p_{SFP}} \cdot Tf_t(1 - p_{SFN} - p_{SFP}) \\
&= f_t,
\end{aligned}
\tag{27}
$$

so that the expected value of $\hat{f}$ is the true value that underlies the testing process.

**The mean of $\hat{\rho}$.**  We begin with Eq 21, rewritten as

$$
\hat{\rho} = \frac{(1 - \hat{f} - \hat{g})}{(1 - p_{AFN} - p_{AFP})\hat{f}} \left[ \frac{P - P_S - p_{AFP}(T - T_S)}{T - T_S} \right].
\tag{28}
$$

Eq 28 is a nonlinear function of $\hat{f}$ and $\hat{g}$ and involves the quotients of the random variables. We can approximate the expectation of $\hat{\rho}$ using the delta-method [30, 49], which involves Taylor expansion of the right hand side of Eq 28 to second order and then taking expectations. (Details are in the S2 Section in S1 File). The result is

$$
\mathcal{E}(\hat{\rho}) = \rho_t + \frac{1}{2} \left[ \frac{2\rho_t}{f_t^2} Var(\hat{f}) - \frac{2}{Tc_2 f_t^2} Cov(\hat{f}, P - P_S) + \frac{2p_{AFP}}{Tc_2 f_t^2} Cov(\hat{f}, T - T_S) \right].
\tag{29}
$$

where $Var(X)$ and $Cov(X, Y)$ denote the variance and covariance of random variables $X$ and $Y$, which arise from the second order Taylor expansion.

The right side of Eq 29 shows that the leading term in the expected value of $\hat{\rho}$ is the true value generating the data and that this is corrected by variances and covariances that account for the nonlinearity in Eq 24.

## Joint properties of $\hat{f}$ and $\hat{\rho}$ via likelihood analysis

Eq 2 for the test range can be obtained by direct manipulation of the relevant random variables [30]. When we separate symptomatic and asymptomatic infections, the compatibility interval for the incidence rate is replaced by a compatibility region (CR) for the incidence rate of symptomatic infected individuals and the ratio of asymptomatic to symptomatic individuals. Because Eqs 18 and 19 are nonlinear in the test results (which are random variables) and Eq 21 is also nonlinear in $\hat{f}$, the analytical approach used in Mangel and Brown [30] is less feasible now.

We develop the analogue of Eq 2 for test range by using likelihood analysis [50–52], exploiting the general property that for a smooth and well-behaved likelihood (which those that follow are), a 95% CR can be approximated by finding the range of variables for which the log-likelihood is below the peak log-likelihood by 1.96 times the number of free parameters. This is essentially a generalization of the Gaussian approximation to the binomial distribution [53] that leads to Eq 2 [30].

We denote the test results by $\tilde{T}_S$, $\tilde{P}_S$, and $\tilde{P}$. For any values of $f$, $g$, and $\rho$, the rules of conditional probability imply (suppressing the dependence on $T$ which is known)

$$
\begin{aligned}
\mathcal{P}(T_S, P_S, P|f, g, \rho) &= \Pr[\tilde{T}_S = T_S, \tilde{P}_S = P_S, \tilde{P} = P|f, g, \rho] \\
&= \Pr[\tilde{T}_S = T_S|f, g] \bullet \Pr[\tilde{P}_S = P_S|T_S, f, g] \bullet \Pr[\tilde{P} = P|P_S, T_S, f, g, \rho].
\end{aligned}
\tag{30}
$$

Each term on the right side of Eq 30 is a binomial distribution. In particular, for any values of $f$, $g$, and $\rho$,

$$\Pr[\tilde{T}_S = T_S | f, g, \rho] = \mathcal{B}(T_S, T, f + g), \tag{31}$$

$$\Pr[\tilde{P}_S = P_S | T_S, f, g, \rho] = \mathcal{B}\left(P_S, T_S, \frac{f(1 - p_{SFN}) + g p_{SFP}}{f + g}\right), \text{ and} \tag{32}$$

$$\Pr[\tilde{P} = P | P_S, T_S, f, g, \rho] = \mathcal{B}\left(P - P_S, T - T_S, \frac{f\rho(1 - p_{AFN}) + (1 - g - f(1 + \rho)) p_{AFP}}{1 - f - g}\right), \tag{33}$$

the probabilities on the right side of Eqs 31–33 are, respectively, $p_S$, $p_{S+}$, and $p_{A+}$ in Eqs 12–14 for any values of $f$, $g$ and $\rho$, rather than their true but unknown values.

When data $T_S$, $P_S$, and $P$ are obtained, the likelihoods, given the data, that the state of the environment is $f$, $g$, and $\rho$ are

$$\mathcal{L}_{T_S}(f, g | T_S, T) = \mathcal{B}(T_S, T, p_S(f, g)), \tag{34}$$

$$\mathcal{L}_{P_S}(f, g | P_S, T_S, T) = \mathcal{B}(P_S, T_S, p_{S+}(f, g)), \text{ and} \tag{35}$$

$$\mathcal{L}_P(f, g, \rho | P, P_S, T_S, T) = \mathcal{B}(P - P_S, T - T_S, p_{A+}(f, g, \rho)). \tag{36}$$

The likelihood of the data $\{T_S, P_S\}$ from symptomatic individuals only depends on the values of $f$ and $g$ and is

$$\mathcal{L}_S(f, g | P_S, T_S, T) = \mathcal{L}_{P_S}(f, g | P_S, T_S, T) \cdot \mathcal{L}_{T_S}(f, g | T_S, T). \tag{37}$$

and the total likelihood of all the data $\{T_S, P_S, P\}$ is

$$\mathcal{L}(f, g, \rho | P, P_S, T_S, T) = \mathcal{L}_P(f, g, \rho | P, P_S, T_S, T) \cdot \mathcal{L}_S(f, g | P_S, T_S, T). \tag{38}$$

The likelihoods in Eqs 37 and 38 are products of binomial distributions that are well approximated, for sufficient numbers of tests, by the appropriate Gaussian distribution [30, 53]. In the results, we will explore log-likelihoods for both the binomial distributions and their Gaussian approximations.

**Simplifying the likelihoods.** Keeping our eyes on the prize of computing the risk of including infected but asymptomatic individuals in groups of different sizes, we focus on $f$ and $\rho$ when constructing the CR. Exploring the likelihood is more convenient if one can eliminate having to deal with $g$ explicitly. Two methods are the profile likelihood and the marginal likelihood [49]; both reduce the number of parameters from 3 to 2.

For the profile likelihood, we replace $g$ in Eqs 37 and 38 by the MLE $\hat{g}$, so that

$$\mathcal{L}_{S, profile}(f | P_S, T_S, T) = \mathcal{L}_{P_S}(f, \hat{g} | P_S, T_S, T) \cdot \mathcal{L}_{T_S}(f, \hat{g} | T_S, T) \text{ and} \tag{39}$$

$$\mathcal{L}_{profile}(f, \rho | P, P_S, T_S, T) = \mathcal{L}_P(f, \hat{g}, \rho | P, P_S, T_S, T) \cdot \mathcal{L}_S(f, \hat{g} | P_S, T_S, T). \tag{40}$$

For the marginal likelihood, we integrate Eqs 37 and 38 over $g$, so that

$$\mathcal{L}_{S,marginal}(f|P_S, T_S, T) = \int_0^1 \mathcal{L}_{P_S}(f, g|P_S, T_S, T) \cdot \mathcal{L}_{T_S}(f, g|T_S, T)dg \text{ and} \tag{41}$$

$$\mathcal{L}_{marginal}(f, \rho|P, P_S, T_S, T) = \int_0^1 \mathcal{L}_P(f, g, \rho|P, P_S, T_S, T) \cdot \mathcal{L}_S(f, g|P_S, T_S, T)dg. \tag{42}$$

By numerical exploration, I found that for the operational questions modeled here, the two methods give virtually the same results for the answers. Were we interested in the tails of the likelihood, this might not be the case. Since the profile likelihood is computationally much speedier, I report results using it. The third Rscript in S4 Section in S1 File allows one to explore the differences between marginal and profile likelihoods for the symptomatic data.

**The compatibility regions from the profile likelihood.** I computed the approximate 95% CR from the total profile likelihood using a generalization of the method of Hudson [51] by first finding the maximum value of the profile log-likelihood and then determining the region in $f$, $g$, or $f$, $\rho$-space in which the log-likelihood was $2 \cdot 1.96 = 3.92$ below its maximum value.

I did computations using R Studio 1.0.143 with underlying R 3.6.1 GUI 1.70 El Capitan build (7684) on an iMac running Mac OS 12.1.

## Results

### Test range with no information on symptoms and positivity less than the probability of a false positive test

When positivity is less than the probability of false positive test, $\varphi(f|P, T)$ has, similar to the likelihood, its maximum at $f = 0$ and monotonically declines. The peak value of $\varphi(f|P, T)$ and the rate of decline depend on the positivity and the number of tests (Fig 5).

These normalized likelihoods In Fig 5 have a test range that depends on the number of tests (Fig 6). As with the situation in which positivity exceeds the probability of a false positive test, the test range declines with test numbers but at a decreasing rate.

### Point estimates, compatibility regions, and risk when there is information on symptoms

In the base case for Monte Carlo simulations, I set $N = 1000$ replicates of $T = 1500$ tests. Since simulation and test errors scale as the reciprocal of their values, these choices have inherent errors of the order of 3%, which are sufficient to understand the qualitative patterns and most of the quantitative patterns. I chose the parameters for the true state of the world and the test errors from those reported in [22–27]: $f_t = 0.05$, $g_t = 0.04$, and $\rho_t = 1.5$ and the test errors are $p_{SFN} = 0.25$, $p_{SFP} = 0.03$, $p_{AFN} = 0.5$, and $p_{AFP} = 0.003$. S3 Section in S1 File contains results for other choices of the true but unknown state of the world.

For the likelihood calculations and the associated risk computations, I first assume that the test results are the expected values $T_S$, $P_S$, and $P$, which is a reasonable assumption when $T$ is large enough, after which I allow the test results to vary more widely. For the base case parameters, the mean values are $\bar{T}_S = 135$, $\bar{P}_S = 58.05$, and $\bar{P} = 118.0575$. Since actual test data can only produce integer values, I rounded the $\bar{P}_S$ and $\bar{P}$ to 58 and 118, respectively. Doing so gives the point estimates $\hat{f} = 0.04995$, $\hat{g} = 0.04005$, and $\hat{\rho} = 1.50119$ (significant digits included to illustrate how little accuracy is lost by the rounding process; since the true values are $f_t = 0.05$, $g_t = 0.04$, and $\rho_t = 1.5$).

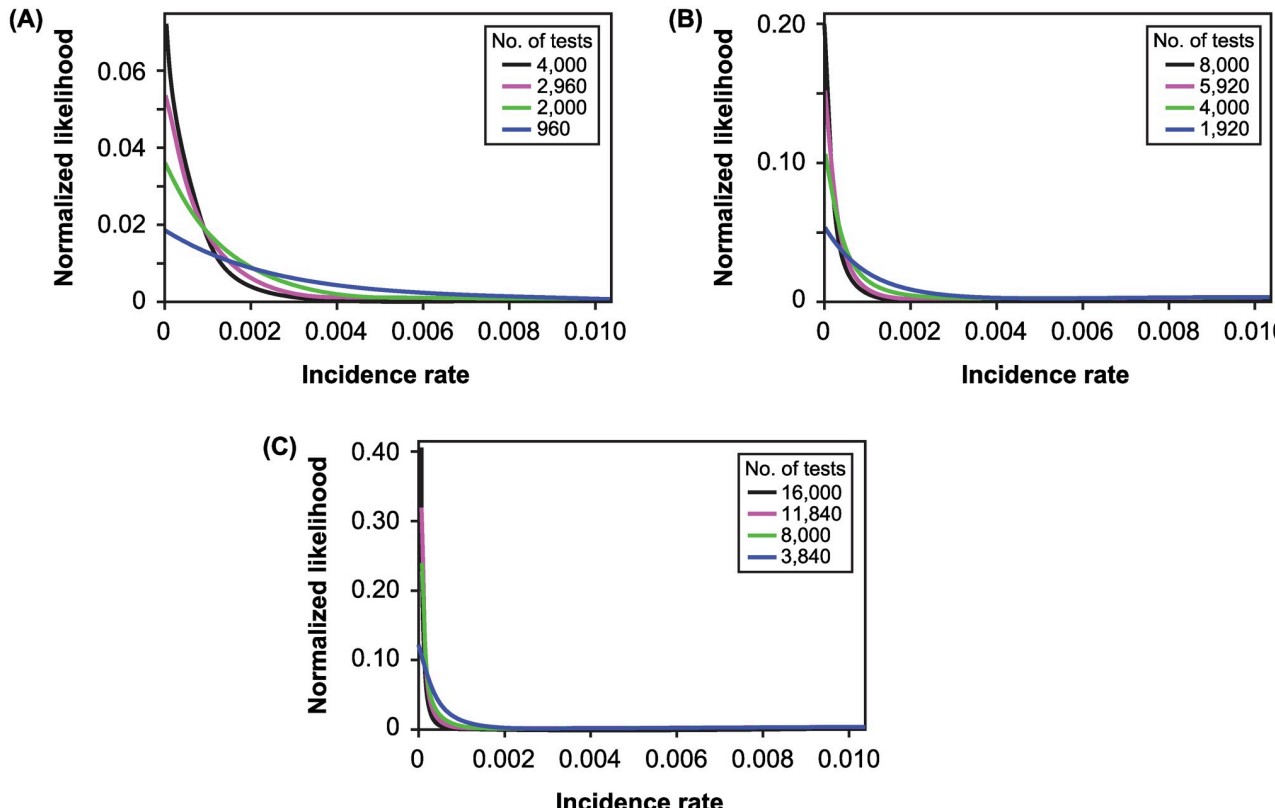

**Fig 5. The normalized likelihoods for incidence rate.** Shown are normalized likelihoods (i.e., posteriors with a uniform prior) when $p_{FN} = 0.25$, $p_{FP} = 0.05$, and positivity is (A) 0.025, (B) 0.0125, or (C) 0.00625. The colored curves correspond to different numbers of tests shown in the legend inset; since positivity is specified, higher numbers of tests are associated with lower levels of positivity.

**Illustrative simulated data.** The $n^{th}$ replicate of the simulation of the testing process yields estimates $\hat{f}_n$, $\hat{g}_n$, and $\hat{\rho}_n$. In Fig 7, I show the first 100 values of the simulation replicates. Each circle represents the value of $\hat{f}_n$, $\hat{g}_n$, or $\hat{\rho}_n$ on the $n^{th}$ replicate of the simulation. The thick red lines represent the averages over the entire 1000 simulations. There are also black lines at the true values of the three parameters.

The means of $\hat{f}_n$ and $\hat{g}_n$ essentially sit on top of the true values, as we would expect from the analysis in Eqs 22–27 showing that $\mathcal{E}(\hat{f}) = f_t$ and $\mathcal{E}(\hat{g}) = g_t$. To quantify this agreement, I computed the mean relative error (ME) for the three estimates. For example, for $\hat{f}$, it is

$$ME(\hat{f}_n) = \frac{\frac{1}{N}\sum_{n=1}^{N}\hat{f}_n - f_t}{f_t}. \tag{43}$$

For the simulation illustrated in Fig 7, $ME(\hat{f}_n) = 0.0039$ and $ME(\hat{g}_n) = 0.0036$ (i.e., both a fraction of a percent).

The lower right panel of Fig 7 has an expanded y-axis to show that the mean of the $\hat{\rho}_n$ exceeds $\rho_t$. For this run of the simulation, $ME(\hat{\rho}_n) = 0.0141$. While this is less than 2.0%, it is almost four times larger than the mean errors of $\hat{f}_n$ and $\hat{g}_n$.

In Fig 7, the thin dotted lines show the means of the estimates ±1.96 times their standard deviations. These are a naive 95% compatibility interval under a Gaussian approximation

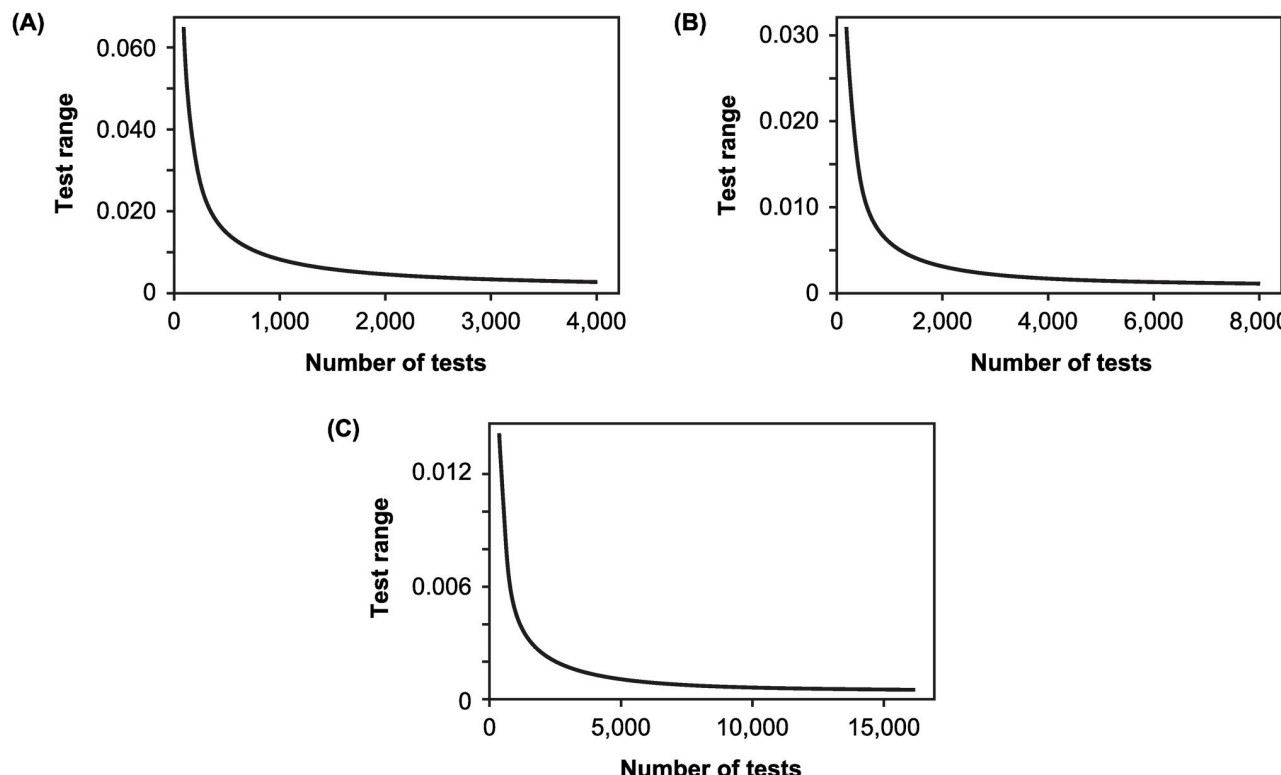

**Fig 6. The test range for incidence rate.** Shown are the test ranges for posteriors with a uniform prior when $p_{FN}$ = 0.25, $p_{FP}$ = 0.05, and positivity is (A) 0.025, (B) 0.0125, or (C) 0.00625.

because they ignore the other two parameters. Even so, for the full set of 1000 replicates of the simulation, the fractions of points outside this naive interval are 0.045, 0.055, and 0.05, respectively, for $\hat{f}_n$, $\hat{g}_n$, and $\hat{\rho}_n$.

We conclude that the formulas for the MLEs accurately capture their true values. The specific results, of course, depend on the simulation results and the number of tests given (both addressed in the next section). For example, in a different run of 1000 simulations of the testing process, the mean relative errors were -0.0073, -0.0006, and -0.033 for $\hat{f}_n$, $\hat{g}_n$, and $\hat{\rho}_n$ respectively, and the fractions of points outside of the naive 95% CR were 0.054, 0.035, and 0.045 for $\hat{f}_n$, $\hat{g}_n$, and $\hat{\rho}_n$, respectively.

**Likelihood, compatibility regions, and the risk of groups of different sizes.** In order to focus on a single value of "test data" we continue using the expected values of $T_S$, $P_S$, and $P$. After exploring the situation when test data are the mean value, we will vary the test data.

**The likelihood of the symptomatic data.** On the way to the goal of estimating the fraction of asymptomatic infections, it is worthwhile to briefly stop and explore the likelihood of the symptomatic data, which are independent of $\rho$ (Eq 37). In Fig 8, I show the likelihood when the means of $T_S$ and $P_S$ are the test results. In this figure, the white dot denotes the true values of parameters and sits at the peak of the heat map.

When the incidence rate is $f$ and the fraction of symptomatic individuals who are uninfected is $g$, the mean value of $T_S = T(f + g)$ so that we expect a negative correlation between values of $f$ and $g$, which is evidenced in the figure by the orientation of the contours of likelihood.

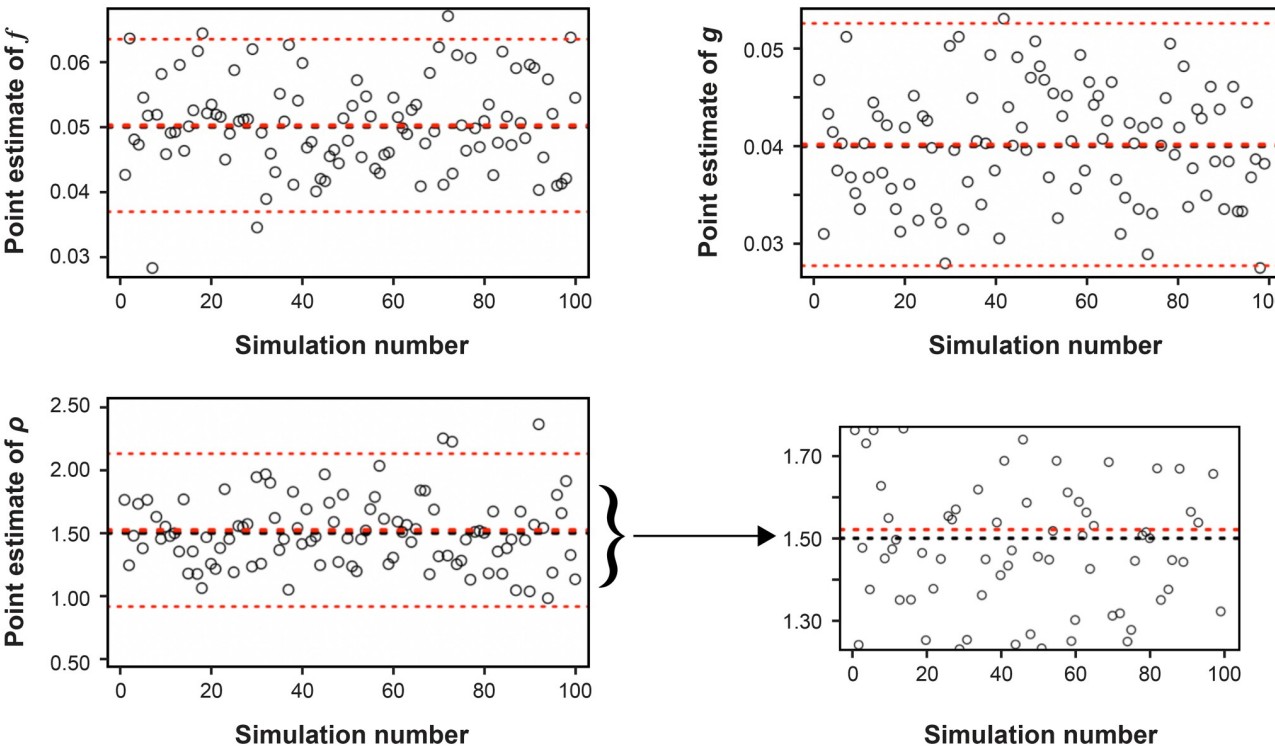

**Fig 7. Results of simulating the process of testing.** Shown (for ease of presentation) are the first 100 values of the point estimates for $f_t$ (upper left panel), $g_t$ (upper right panel), and $\rho_t$ (lower left panel). The lower right panel is an expanded version of the point estimates for $\rho_t$. Each circle represents the value of $\hat{f}_n$, $\hat{g}_n$, or $\hat{\rho}_n$ on the $n^{th}$ replicate of the simulation. The thick red lines represent the averages over the entire 1000 simulations. There are also black lines at the true values of the three parameters. In the lower right panel, the y-axis is expanded to show that the mean of the $\hat{\rho}_n$ exceeds $\rho_t$; see the text for an explanation. The means of $\hat{f}_n$ and $\hat{g}_n$ essentially sit on top of the true values; again see the text for an explanation. The thin dotted lines show the means of the estimates ±1.96 times their standard deviations.

For the purposes of the risk calculation, the most important role of the likelihood function of the symptomatic data is to provide the MLE value of $g$ for construction of the profile likelihood in Eq 40, to which we now turn.

**The likelihood of all the data.** In Fig 9, I show the profile likelihood (Eq 40) for $f$ and $\rho$ when the test data are the mean values of of $T_S$, $P_S$, and $P$. The banana shape of the contours of the 95% CR computed is a result of the nonlinearity in Eq 28. In this case, the likelihood is centered at the true values of the parameters (shown by the white dot), and when Eq 28 is converted to a function $\rho(f)$ by using the MLE $\hat{g}$ and replacing $\hat{f}$ by $f$ and $\hat{\rho}$ by $\rho$, the true values of the parameters sit on the resulting curve, which runs through the middle of the 95% CR.

One property of the range formula in Eq 2 is that test range declines as $1/\sqrt{T}$. That is, although the range declines as the number of tests increases, it does so at a decreasing rate [30, Figures 6-8 and p. 16ff]. This observation is more than an academic point, because it has the operational implication that it is possible to over-sample by providing too many tests in a single spatial region (also [30]).

In Fig 10, I explore the consequences of simultaneously increasing the number of tests and relaxing the assumption that the test data are the mean values of $T_S$, $P_S$, and $P$ so that the true values (the white dots) no longer sit in the middle of the 95% CR or on the curve $\rho(f)$ and the contours move in space, as determined by the test results. As with Eq 2, contours shrink as the number of tests increases, but at a decreasing rate.

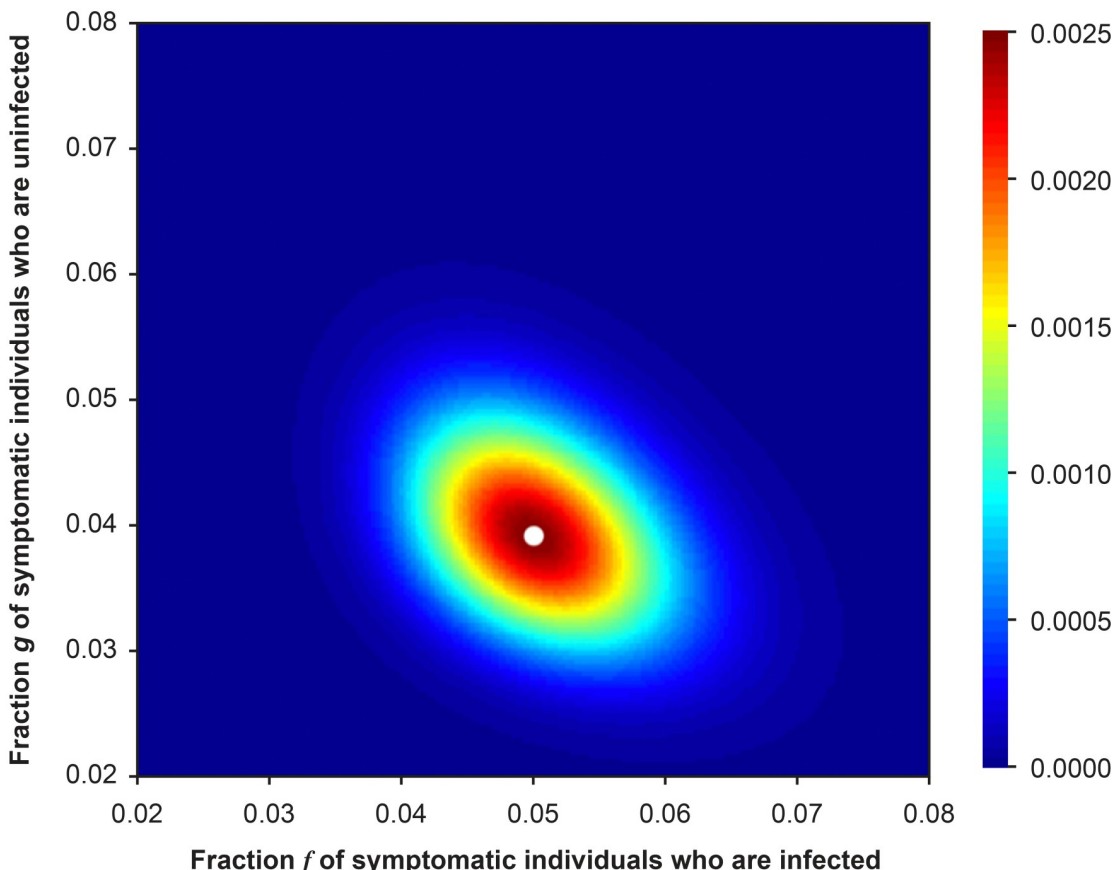

**Fig 8. The likelihood (Eq 37) of the fractions of individuals who are infected and symptomatic, *f*, and uninfected and symptomatic *g*, when the test data are the means of $T_S$ and $P_S$.** The white dot denotes the true values of the parameters.

**The prize: The risk of including asymptomatic infected individuals in groups of different sizes.** We are now able to compute the risk of including asymptomatic but infected individuals in groups of different sizes, to generate a curve analogous to Fig 1. For any values of *f* and *ρ*, the fraction of asymptomatic infected individuals in the population is *ρf*. Hence, the analogue of Eq 3 is

$$\mathcal{R}(h, f, \rho) = 1 - (1 - \rho f)^h. \tag{44}$$

In Fig 11, I show the risk computed using the mean test data, profile likelihood and $\rho f = \hat{\rho}\hat{f}$, the minimum of *ρf* on the 95% CR contour, or the maximum of *ρf* on the 95% CR contour in Eq 44. Clearly, one can invert Eq 44 in analogy to Eq 4 and compute analogues of the results shown in Fig 2.

## Discussion

It is important to recognize that the analysis presented in this paper is a procedure that allows one to go from test information to the risk of including infected individuals in groups of various sizes when there is no information on symptoms at the time of testing or to the risk of including asymptomatic individuals in groups of different sizes when there is information on testing. Rather than being binary (risky or not), this risk is graded and the specific details of

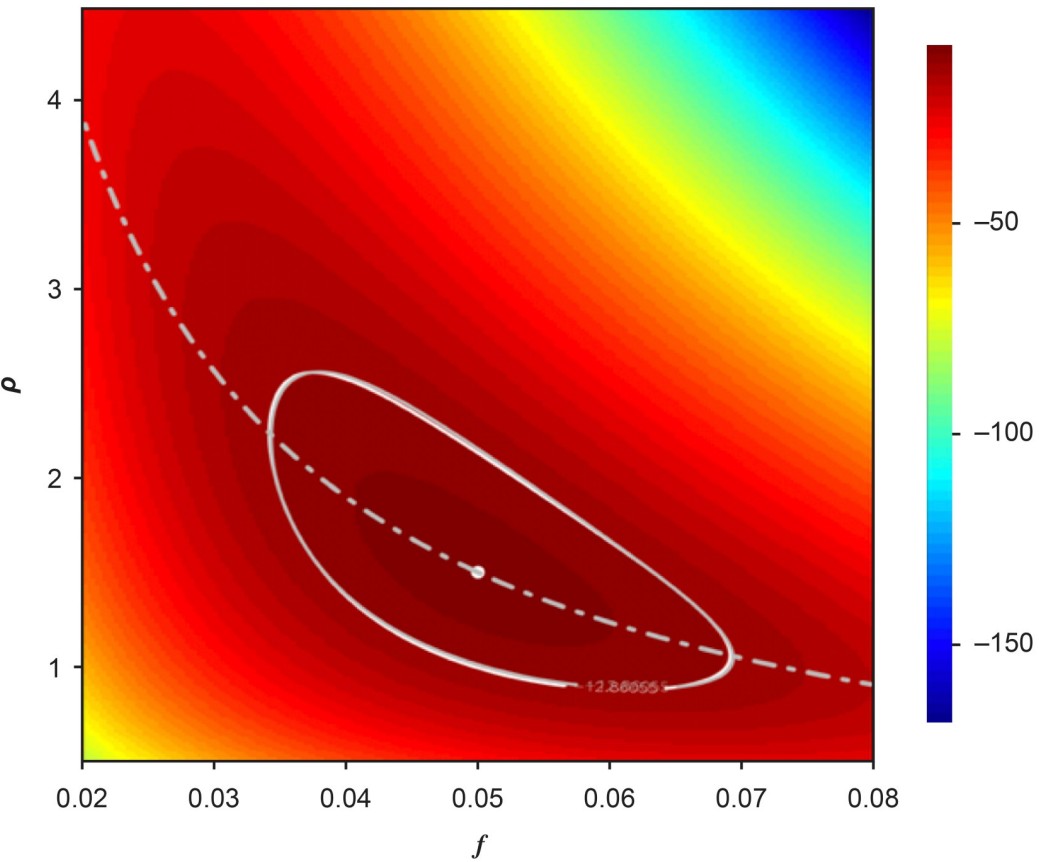

**Fig 9. The profile log-likelihood (Eq 40) for the fraction of individuals who are infected and symptomatic $f$ and the ratio of of the fraction of individuals who are infected and asymptomatic to those who are infected and symptomatic $\rho$ when the test data are the means of $T_S$, $P_S$, and $P$.** The white dot denotes the true values of the parameters. The 95% CR contour is shown in white for the exact binomial likelihoods and in gray for the Gaussian approximation to those likelihoods. The white dotted line is obtained by replacing $\hat{g}$ in Eq 28 by its MLE value, replacing $\hat{f}$ by an arbitrary value $f$, and then viewing the right side as an equation for the ratio $\rho(f)$ of asymptomatic to symptomatic infected individuals.

the relationship between group size and risk depends upon the operational details of testing such as test numbers and errors. Once these are specified, the procedures can be employed.

Let us now consider three limitations of the methods developed here. First note that Eq 26 has the same problem as Eq 1 when the positivity is very small. To see this, we factor out $T_S$ on the far right side of Eq 26 to obtain

$$\hat{f} = \frac{T_S}{T}\left[\frac{P_S/T_S - p_{SFP}}{1 - p_{SFN} - p_{SFP}}\right],$$

so that if the positivity rate among symptomatic individuals falls below the probability of a false positive test among symptomatic individuals, $\hat{f}$ is less than zero. As in the situation with no information on symptoms, the operational interpretation is that we then set $\hat{f}$ to 0. Alternatively, we may generalize the analysis for the simpler case by putting a prior on the parameters and determining the CR in that manner.

Second, an objection may be made that the binomial distribution underlying the analysis relies on the strong assumption that tests are independent events but that often groups of people will test together so that modeling the testing process requires an aggregated distribution.

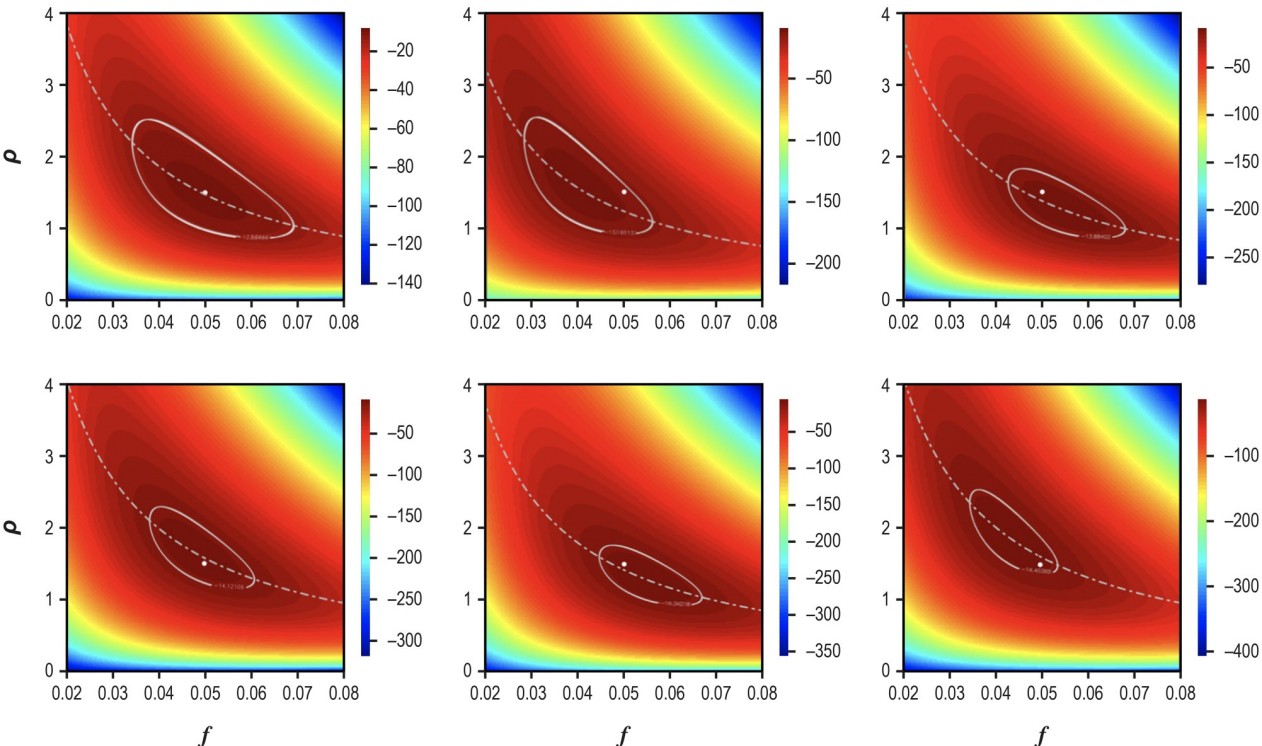

**Fig 10. The consequences of varying test numbers and letting test data vary from the mean values of $T_S$, $P_S$ and $P$.** In each panel the white dot represents $f_t$ and $\rho_t$, and the dotted curve is the function $\rho(f)$ described in the caption to the previous figure and which now depends on the test results. The upper left panel reproduces Fig 9, in which $T = 1500$ and the test data are the mean values of $T_S$, $P_S$, and $P$. In the other panels, the test data are a random realization of the simulation of the testing process and going clockwise from the upper left panel, $T = 2000, 3000, 3500, 4000$, and $4500$.

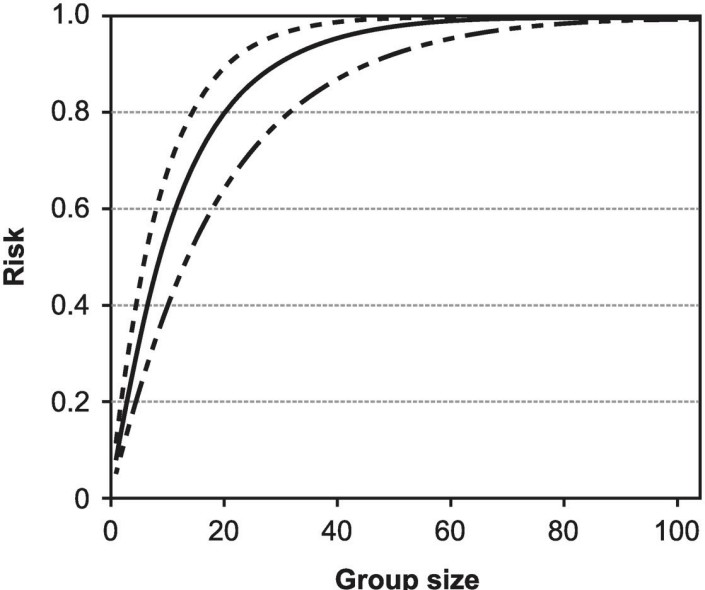

**Fig 11. The risk of including asymptomatic individuals in groups of different sizes.** The solid line corresponds to using the MLEs for $\hat{f}$, $\hat{g}$, and $\hat{\rho}$ in the risk formula (Eq 44), and the two dotted lines correspond to using the minimum and maximum values of $\rho f$ on the 95% CR contour.

This is a fair objection, however: 1) the binomial distribution is the appropriate starting point, and if the sample is large and diverse enough (e.g., from many different testing sites), the independence assumption should be at least approximately valid; and 2) a negative binomial distribution of the form used in ecology to model aggregated counts [48, pp. 103–111] is a natural starting point for extending the work here.

Third, an objection may be made that we have assumed the values for test errors rather than estimating them. DiCiccio et al. [54] show that estimating test errors at the same time as incidence rate is a much more complex problem, and is likely one whose solution is not easily transferred to recommendations for practice. An alternative is to stratify test results by both symptomatic or not and time since putative exposure, have approximate values for the test errors for each time since exposure, and conduct sensitivity analysis by varying the values of the test errors. Furthermore, Mangel and Brown [30, pp. 25–27] show how to generalize Eq 1 for the case of a distribution of test errors using the delta-method. A similar extension of Eqs 18, 19 and 21 is a potential next step in this work.

In some locations, individuals are already asked whether they are symptomatic or not at the time of testing. For example, Nomi Health in Utah requires a self-reporting form for obtaining a coronavirus test, and the form includes yes or no questions such as: "Do you have a fever, a cough, new or increased shortness of breath, decreased smell or taste, a sore throat, muscle aches or pains, a headache, congestion or a runny nose, nauseas or vomiting, diarrhea, fatigue?"

During the 2020–2022 academic years, natural experiments in testing were occurring on college campuses [55]. The results of those tests will provide a trove of information to explore with the methods developed here.

## Conclusions

In conclusion, the approach of modeling and simulating the process of testing before analyzing testing data leads to a range of insights and at least the following operational recommendations:

- At the time of testing, collect information on whether and individual is symptomatic or not.

- At the time of testing, collect information on putative time since exposure to infection.

- Conduct experiments to obtain information on means and variances of test errors.

  There is much to be done and no time to lose before the next pandemic.

## Supporting information

**S1 File. Including a brief review of the binomial distribution and likelihood, a mathematical appendix with details of calculations in the main text, sensitivity analysis when there is information on symptoms, and codes that generate the results in the main text and sensitivity analysis.**
(PDF)

## Acknowledgments

I thank Alan Brown for inviting me, early in the pandemic, to think about operational analysis of testing for coronavirus with him and Matt Shaffer for his steady encouragement during the work. For comments on presentations and this manuscript, I thank three anonymous referees, Tiffany Bogich, Rebecca Borchering, Alan Brown, Emily Howerton, John Ivancovich, Elyse Johnson, Chaya Pflugeisen, Matt Shaffer, Katriona Shea, and Joseph Travis.

## Author Contributions

**Conceptualization:** Marc Mangel.

**Methodology:** Marc Mangel.

**Software:** Marc Mangel.

**Visualization:** Marc Mangel.

**Writing – original draft:** Marc Mangel.

**Writing – review & editing:** Marc Mangel.

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
