## [Decision Letter · Decision Letter 0]

26 Oct 2022

PONE-D-22-20932Operational analysis for coronavirus testing: Positivity is not incidence, risk is not binary, and the fraction of asymptomatic infections can be determinedPLOS ONE

Dear Dr. Mangel,

Thank you for submitting your manuscript to PLOS ONE. After careful consideration, we feel that it has merit but does not fully meet PLOS ONE’s publication criteria as it currently stands. Therefore, we invite you to submit a revised version of the manuscript that addresses the points raised during the review process.

We look forward to receiving your revised manuscript.

Kind regards,

Rehana Naz

Academic Editor

PLOS ONE

Journal Requirements:

. When submitting your revision, we need you to address these additional requirements.

"Financial support was provided by a consulting agreement with the Johns Hopkins University Applied Physics Laboratory (APL). I thank Alan Brown for inviting me, early in the pandemic, to think about operations analysis of testing for coronavirus and Matt Shaffer for his steady support during the work."

"MM was supported by a consulting contract with the Johns Hopkins University Applied Physics Laboratory. The funders had no role in study design, data collection and analysis, decision to publish, or preparation of the manuscript."

Reviewers' comments:

Reviewer's Responses to Questions

**Comments to the Author**

1. Is the manuscript technically sound, and do the data support the conclusions?

Reviewer #1: Yes

Reviewer #2: Partly

Reviewer #3: Yes

 2. Has the statistical analysis been performed appropriately and rigorously?

Reviewer #1: Yes

Reviewer #2: No

Reviewer #3: Yes

 3. Have the authors made all data underlying the findings in their manuscript fully available?

Reviewer #1: Yes

Reviewer #2: Yes

Reviewer #3: Yes

 4. Is the manuscript presented in an intelligible fashion and written in standard English?

Reviewer #1: Yes

Reviewer #2: Yes

Reviewer #3: Yes

 5. Review Comments to the Author

Reviewer #1: The author has shown new applications of a method to translate surface positivity to estimate incidence rates. The author has also discussed limitations of the study. The manuscript is well written and is a significant contribution to the field. I recommend it for publication.

Reviewer #2: The following points should be considered in the manuscript:

1-In the equation-6, the relationship is proportional to and how the author took the logarithm and the derivative of the two side themselves in the equation-7 and even for the far right side, the details must be explained in the manuscript.

2-In the figures-8, 9, and 10, the results give high possibilities of testing errors, of course based on the author results, however, this is not the actual case and the author must add lots of details about those results and their relation to the actual cases.

3-The title of the manuscript is a vague and only can be used in blogs rather than in scientific article and this title must be reworded.

4-There are some equations are approximated equations, especially, in thé appendices. So, the author must mention this point in the manuscript for each one.

5-There are some symbols or abbreviations used in the manuscript without defining them, so, the author must define each symbol even if the symbol is a well known.

6-It is not preferable at all using the first person pronoun 'I' as the author did in the manuscript, especially, in the abstract.

7-The reasons of choosing of the value of the (MLE) fˆ in the results of the figures must be explained in details.

8-The author in the title of the manuscript and other places of the manuscript has used the term 'coronavirus', however, the manuscript focuses on a specific type of the coronaviruses and this point must be reconsidered.

9-The computational errors of the calculations must be discussed with lots of details because the results of the fractions are not consistent with the actual case.

Reviewer #3: The author summarizes some results from a prior study, where an estimator is derived for the incidence rate if the surface positivity and probabilities of false positive and false negative tests are known and the range for the value of the incidence rate (‘test rate’) is obtained. A formula for the test range is derived in the case when the surface positivity is less than the probability of a false positive. Further, the results of the earlier study are generalized when information regarding symptomatic and asymptomatic individuals is known at the time of testing, including point estimate and range for the ratio of asymptomatic to symptomatic cases.

Comments:

The work deals with a very relevant problem, and is well communicated. Possible concerns regarding the approach have been spelt out and satisfactorily answered in the ‘Discussion’ section.

I would suggest that since the work draws from the previous reports ([29] and [30]), it may be a good idea to include the calculations from those studies in the mathematical appendix, as this will make this paper more self-contained.

There may be a typo on Line 281

 6. PLOS authors have the option to publish the peer review history of their article (what does this mean?). If published, this will include your full peer review and any attached files.

Reviewer #1: No

Reviewer #2: No

Reviewer #3: **Yes**

---

## [Author Response · Author response to Decision Letter 0]

16 Dec 2022

Response to comments from the reviewers

1. Is the manuscript technically sound, and do the data support the conclusions?

Reviewer #1: Yes

Reviewer #2: Partly

Reviewer #3: Yes

Response: In response to the specific comments of Reviewer #2, detailed below, I addressed all the points that lead to the “Partly” answer.

 2. Has the statistical analysis been performed appropriately and rigorously?

Reviewer #1: Yes

Reviewer #2: No

Reviewer #3: Yes

Response: The statistical analysis, both Bayesian and likelihood methods, is indeed rigorously performed. I believe that the answer from Reviewer #2 is a result of the Reviewer expecting more details of some of the analytical steps. As described below, these are now included, both in the main text ant the supplementary information (depending upon the level of detail involved).

 3. Have the authors made all data underlying the findings in their manuscript fully available?

Reviewer #1: Yes

Reviewer #2: Yes

Reviewer #3: Yes

Response: No response needed.

 4. Is the manuscript presented in an intelligible fashion and written in standard English?

Reviewer #1: Yes

Reviewer #2: Yes

Reviewer #3: Yes

Response: No response needed. 

 5. Review Comments to the Author

Reviewer #1: The author has shown new applications of a method to translate surface positivity to estimate incidence rates. The author has also discussed limitations of the study. The manuscript is well written and is a significant contribution to the field. I recommend it for publication.

Response: Thank you for these comments. Although the manuscript organization has not changed, I worked to improve and tighten the writing and, as detailed below in the response to Reviewer #2, I details of analysis at various points in the manuscript. These are indicated by the line numbers in the new version of the manuscript.

Reviewer #2: The following points should be considered in the manuscript:

Response: Thank you for these thoughtful comments, which helped me make the approach clearer.

1-In the equation-6, the relationship is proportional to and how the author took the logarithm and the derivative of the two side themselves in the equation-7 and even for the far right side, the details must be explained in the manuscript.

Response: I removed the proportional to in Eqn 6, using equality instead and then added a sentence below it explaining that although the mathematical equation is the same, the variables play different roles. I also expanded Supplementary information S1 Brief review of the binomial distribution and binomial likelihood to include a new sub-section that contains a derivation of Eqn 7. 

I believe that the detailed derivation is better left in the supplementary information because it is possible to then include all of the mathematical details, taking nothing for granted on the part of the reader. That is, the mathematical tools are relative simple but used in mature ways and putting the derivation in the supplementary information allows me to be very explicit and clear about how the derivation works.

2-In the figures-8, 9, and 10, the results give high possibilities of testing errors, of course based on the author results, however, this is not the actual case and the author must add lots of details about those results and their relation to the actual cases.

Response: I now cite references 22-27 to justify the choice of the true state of nature and the test errors (lines 316-319) and make reference to supplementary information S3 Sensitivity analysis when there is information on symptoms for other assumptions about the true state of nature. 

 In addition, I added a point in the discussion (lines 403-410) to emphasize that what is developed here is a procedure to go from test information to the risk of including asymptomatic individuals in groups of different sizes, so that one can apply the procedure to their own choice of true states of nature and test errors.

3-The title of the manuscript is a vague and only can be used in blogs rather than in scientific article and this title must be reworded.

Response: As the reviewer surmised, I struggled with finding the right title. The title is now

“ Operational analysis for COVID-19 testing: Determining the risk from asymptomatic infections”

4-There are some equations are approximated equations, especially, in thé appendices. So, the author must mention this point in the manuscript for each one.

Response: The approximated equations are explained in more detail (lines 240-246 and below Eqn S19 in the Supporting information), along with additional citation to reference 30 in which the delta method is explained in detail.

5-There are some symbols or abbreviations used in the manuscript without defining them, so, the author must define each symbol even if the symbol is a well known.

Response: All symbols are now explained at the very first time they are introduced.

6-It is not preferable at all using the first person pronoun 'I' as the author did in the manuscript, especially, in the abstract.

Response: I removed the first person from the abstract, but after checking with the editor handling the manuscript left first person in some places in the main text, because otherwise I would have had to start writing in passive rather than active tense.

7-The reasons of choosing of the value of the (MLE) fˆ in the results of the figures must be explained in details.

Response: Additional explanations for the choice of MLE (lines 37-39) and the compatibility intervals and regions (lines 46-49, 262-264) are given.

8-The author in the title of the manuscript and other places of the manuscript has used the term 'coronavirus', however, the manuscript focuses on a specific type of the coronaviruses and this point must be reconsidered.

Response: I now refer to COVID-19 in the title and in the first sentence of the manuscript write “Entering the third year of the 2019 coronavirus disease (henceforth COVID-19)

9-The computational errors of the calculations must be discussed with lots of details because the results of the fractions are not consistent with the actual case.

Response: I have added discussion, particularly regarding Figure 10, concerning how we interpret the results of the calculations when they differ from the true state of nature. It is possible, however, that I misunderstand what is meant in this comment by “the actual case”

Reviewer #3: The author summarizes some results from a prior study, where an estimator is derived for the incidence rate if the surface positivity and probabilities of false positive and false negative tests are known and the range for the value of the incidence rate (‘test rate’) is obtained. A formula for the test range is derived in the case when the surface positivity is less than the probability of a false positive. Further, the results of the earlier study are generalized when information regarding symptomatic and asymptomatic individuals is known at the time of testing, including point estimate and range for the ratio of asymptomatic to symptomatic cases.

Comments:

The work deals with a very relevant problem, and is well communicated. Possible concerns regarding the approach have been spelt out and satisfactorily answered in the ‘Discussion’ section.

Response: An excellent summary.

I would suggest that since the work draws from the previous reports ([29] and [30]), it may be a good idea to include the calculations from those studies in the mathematical appendix, as this will make this paper more self-contained.

Response: The two previous reports are easily accessed through the links given in the list of references, so I have not followed this suggestion because it would add considerable length to the paper.

There may be a typo on Line 281

Response: I have rewritten the line, because the explanation of the method is given earlier in the paper. In essence, I am generalizing Hudson’s (1971) method in which the confidence interval (I use compatibility interval) is found by moving 2 units down from the peak of the log-likelihood (I am using 1.96) instead of 2 (lines 295-298) This is the case when there is no information on symptoms so that there is 1 unknown parameter. When there is information on symptoms, there are two unknown parameters so the distance from the peak of the likelihood is twice that of the univariate case.

---

## [Decision Letter · Decision Letter 1]

31 Jan 2023

Operational analysis for COVID-19 testing: Determining the risk from asymptomatic infections

PONE-D-22-20932R1

Dear Dr. Mangel,

We’re pleased to inform you that your manuscript has been judged scientifically suitable for publication and will be formally accepted for publication once it meets all outstanding technical requirements.

Kind regards,

Rehana Naz

Academic Editor

PLOS ONE

Reviewer's Responses to Questions

**Comments to the Author**

1. If the authors have adequately addressed your comments raised in a previous round of review and you feel that this manuscript is now acceptable for publication, you may indicate that here to bypass the “Comments to the Author” section, enter your conflict of interest statement in the “Confidential to Editor” section, and submit your "Accept" recommendation.

Reviewer #1: All comments have been addressed

Reviewer #2: All comments have been addressed

Reviewer #3: All comments have been addressed

2. Is the manuscript technically sound, and do the data support the conclusions?

Reviewer #1: Yes

Reviewer #2: Partly

Reviewer #3: Yes

3. Has the statistical analysis been performed appropriately and rigorously? 

Reviewer #1: Yes

Reviewer #2: Yes

Reviewer #3: Yes

4. Have the authors made all data underlying the findings in their manuscript fully available?

Reviewer #1: Yes

Reviewer #2: Yes

Reviewer #3: Yes

5. Is the manuscript presented in an intelligible fashion and written in standard English?

Reviewer #1: Yes

Reviewer #2: Yes

Reviewer #3: Yes

6. Review Comments to the Author

Reviewer #1: (No Response)

Reviewer #2: The authors have improved lots of points in the revised version of the manuscript, especially the first and the third critical point. The manuscript is more preferable in the revised form.

Reviewer #3: All concerns have been adequately addressed and the updated manuscript may be accepted for publication.

7. PLOS authors have the option to publish the peer review history of their article (what does this mean?). If published, this will include your full peer review and any attached files.

Reviewer #1: No

Reviewer #2: No

Reviewer #3: **Yes: **Adnan Ahmed Khan

---

## [Editor Report · Acceptance letter]

3 Feb 2023

PONE-D-22-20932R1 

Operational analysis for COVID-19 testing: Determining the risk from asymptomatic infections 

Dear Dr. Mangel:

I'm pleased to inform you that your manuscript has been deemed suitable for publication in PLOS ONE. Congratulations! Your manuscript is now with our production department. 

Kind regards, 

on behalf of

Prof Rehana Naz 

Academic Editor

PLOS ONE